# Behavioral and Sleep Disorders in Children and Adolescents following COVID-19 Disease: A Case-Control Study

**DOI:** 10.3390/children10071189

**Published:** 2023-07-10

**Authors:** Michele Miraglia del Giudice, Angela Klain, Giulio Dinardo, Elisabetta D’Addio, Chiara Lucia Bencivenga, Cristina Fontanella, Fabio Decimo, Giuseppina Rosaria Umano, Margherita Siciliano, Marco Carotenuto, Cristiana Indolfi

**Affiliations:** 1Department of Woman, Child and General and Specialized Surgery, University of Campania “Luigi Vanvitelli”, 80138 Naples, Italy; michele.miraglia@unicampania.it (M.M.d.G.); dinardogiulio@gmail.com (G.D.); elisabettadaddio3@gmail.com (E.D.); chiara.bencivenga03@gmail.com (C.L.B.); fabio.decimo@unicampania.it (F.D.); giusi.umano@gmail.com (G.R.U.); cristianaind@hotmail.com (C.I.); 2Clinic of Child and Adolescent Neuropsychiatry, Department of Mental Health, Physical and Preventive Medicine, University of Campania “Luigi Vanvitelli”, 80138 Caserta, Italy; cristina.fontanella@hotmail.it (C.F.); margherita.siciliano@unicampania.it (M.S.); marco.carotenuto@unicampania.it (M.C.)

**Keywords:** COVID-19, children, adolescents, behavior, sleep, neuropsychiatric disorders, long COVID, CBCL, SDSC

## Abstract

Introduction: Recent studies show that neuropsychiatric disorders are the most frequent sequelae of COVID-19 in children. Purpose: Our work aimed to evaluate the impact of SARS-CoV-2 infection on behavior and sleep in children and adolescents. Materials and Methods: We enrolled 107 patients aged 1.5–18 years who contracted COVID-19 between one year and one month prior to data collection, referred to the University of Campania Luigi Vanvitelli in Italy. We asked their parents to complete two standardized questionnaires for the assessment of behavior (Child Behavior CheckList (CBCL)) and sleep (Sleep Disturbance Scale for Children (SLDS)). We analysed and compared the results with a control group (pre-COVID-19 pandemic). Results: In the COVID-19 group, the major results were found for sleep breathing disorders, sleep–wake transition disorders and disorders of initiating and maintaining sleep for the SDSC questionnaire, and internalizing scale, total scale and anxiety/depression for the CBCL questionnaire. The comparison of the CBCL results of the cases with the controls revealed statistically significant differences for the following items: internalizing scale, externalizing scale, somatic complaints, total score, thought problems [(*p* < 0.01)], anxious/depressed problems and withdrawn [(*p* < 0.001)]. Conclusions: COVID-19 has impacted children’s and adolescents’ mental health. Adolescents were the most affected patient group for internalizing problems, including anxiety and depression.

## 1. Introduction

The Coronavirus Disease 2019 (COVID-19) epidemic enormously impacted people’s health and lives. The clinical effects of the novel Coronavirus in children have undergone significant investigation and analysis, but the long-term effects that are associated with increased morbidity are, currently, still under study [1,2]. Acute SARS-CoV-2 infection in children and adolescents runs milder than that in adults. The majority of children with COVID-19 develop a mild form of the disease, with 15–30% of children being infected asymptomatically [3,4,5,6]. However, the low case ascertainment of paucisymptomatic children limits the accuracy of prevalence estimations in countries with widespread and high daily case numbers of COVID-19 [7]. Children with symptomatic COVID-19 infection typically have one or more respiratory symptoms, most frequently fever and cough, similar to those of seasonal respiratory viral infections [5,8]. Although COVID-19 has a favourable prognosis in terms of paediatric age, numerous sequelae have been reported, defined as ‘long COVID’ [9]. Numerous clinical definitions for long or post-COVID syndrome have been offered, since there is no standard nomenclature due to the variability of symptoms and uneven assignment. There is still no unanimous definition of long COVID [10]. One of the most common definitions is the one provided by World Health Organization (WHO). WHO’s definition of long COVID or post COVID-19 condition is ‘symptoms occurring at least three months after probable or confirmed SARS-CoV-2 infection. Symptoms must last for at least two months and cannot be explained by an alternative diagnosis. Symptoms may be new onset following recovery from acute SARS-CoV-2 infection or persist from the initial illness. Symptoms may fluctuate or relapse’ [11]. This definition was developed using a strong, protocol-based technique (Delphi consensus), which involved a large, representative group of patients, caregivers, and other stakeholders from various fields [12]. Other definitions of long COVID have been proposed by other international societies [13,14,15] (Table 1).

According to the literature, long COVID in children is a relevant clinical issue affecting the quality of life and impacting activities of daily living [16,17,18]. According to Lopez et al., the prevalence of long-COVID is about 25% in the pediatric population [18]. Most of the research demonstrated that long COVID in pediatric age concerns the neuropsychiatric sphere, including mood symptoms, fatigue and sleep disorders [18,19,20,21,22]. According to a research brief published by the World Health Organisation (WHO), the prevalence of anxiety and depression surged dramatically by 25% globally in the first year of the COVID-19 pandemic [23]. The aim of our study is to evaluate how COVID-19 has impacted children’s behaviour and sleep by comparing a group of children and adolescents who contracted SARS-CoV-2 infection with a control group (pre-COVID-19 pandemic). 

### Neuropsychiatric Aspects of Long COVID in Children: Underlying Mechanisms

COVID-19 pandemic has impacted young children and teenagers’ mental health [24,25,26]. It is currently unclear what causes SARS-CoV-2 infection’s long-term neuropsychological symptoms [27]. On one hand, the containment procedures, particularly in Italy, were quite strict, especially during the first pandemic wave when there was a total lockdown with the closure of all social events and schools. Most after-school activities for kids and teenagers that traditionally take place outside and in groups were cancelled, along with the schools closing. Children and teenagers were living in a physical state of protracted social isolation, increasing their reliance on social networks [28,29]. Children and adolescents faced specific challenges depending on their stage of life and how both the COVID-19 disease and restrictions impacted them [30,31]. 

According to several studies, SARS-CoV-2 can impact neurological status through different mechanisms, including neuroinflammation, endothelial dysfunction, and damage to blood vessels and neurons [32]. Inflammation and hypoxemia brought on by respiratory distress syndrome may lead to nervous system inflammation, increasing the brain barrier’s permeability as the result of the massive release of inflammatory mediators (chemokines, cytokines). Cognitive performance can be affected by the dysregulation of several neural cell types caused by cytokines, chemokines, and reactive microglia, as well as by the disruption of myelin homeostasis and plasticity, and induction of neurotoxic astrocyte reactivity. Neurovascular dysfunction, such as thrombosis and blood–brain barrier disruption, which results in the leaking of fibrinogen and other pro-inflammatory chemicals, can lead to cerebral inflammation and damage [33]. In several patients with COVID-19, autoimmune encephalitis due to anti-neural autoantibodies has been reported [34]. Other researchers found that latent herpesvirus infections, most notably the Epstein–Barr virus, may reactivate in response to COVID-19, which may then cause further inflammation [35,36]. Neuroinvasive infection rarely occurs through angiotensin-2 conversion enzyme (ACE2) receptor. Viral S spike (S) protein binds to the ACE2 receptor, located on nearly every human organ, as well as endothelial cells and the central nervous system. More specifically, the binding of S protein and neuron’s ACE2 receptor activates the S protein by the action of the serine protease, transmembrane protease serine 2 (TMPRSS2) [37]. SARS-CoV-2 may disrupt the blood–brain barrier by attaching to endothelium’s ACE2 receptors, causing oedema, intracranial hypertension, and virus penetration in the central nervous system [38]. The observation of SARS-CoV-2 viral-like particles in brain capillaries’ endothelial cells, pericytes, and astrocytic processes strongly supports the hematogenous–endothelial neuroinvasion-based concept [39]. According to a UK Biobank study that included brain imaging of the same patients before and after COVID-19 as well as controls, there was less grey matter thickness in the orbitofrontal cortex and parahippocampal gyrus, a general decrease in brain size, and a greater decline in cognitive function in patients after COVID-19 compared to controls, even in non-hospitalized patients [40].

It is still challenging to differentiate between neuropsychological symptoms that are sequelae of SARS-CoV-2 infection versus those that are a result of stress and anxiety, owing to pandemic restrictions.

## 2. Materials and Methods

We enrolled pediatric patients who arrived at the department of woman, child and general and specialized woman of University of Campania ‘Luigi Vanvitelli’ in the period January 2022-January 2023. Patients had to meet the following recruitment criteria: (1) aged between 1.5 and 18 years; (2) confirmed diagnosis of COVID-19 by nasopharyngeal swab; (3) SARS-CoV-2 infection that occurred between one year and one month before patient’s recruitment. We excluded patients with known behavioral and sleep disorders, children with unconfirmed COVID-19 or diagnosed more than one year or less than one month ago. Informed consent was obtained for every patient from his/her parent/caregiver. Control group was used to compare the results of the COVID-19 sample. This consisted of a group of children and adolescents with the same characteristics of case cohort: age, gender, as well as the absence of neuropsychiatric disorders’ diagnosis. The parents of the control group filled in the same questionnaires (CBCL and SDSC) in a period before the COVID-19 pandemic. The same screening method was applied to both groups. The Declaration of Helsinki was adequately addressed, and the study was approved by the ethics committee of the University of Campania Luigi Vanvitelli (register number 0029465).

### 2.1. Assessment Tools

The parent or legal guardian of the test participant answered two standardized questionnaires (Child Behavior Checklist (CBCL) 1.5–5 or CBCL 6–18 and Sleep Disturbance Scale for Children (SDSC).

#### 2.1.1. CBCL: Child Behavior Checklist

CBCL 1.5–5 and CBCL 6–18 is an empirical assessment tool, a parent-report form, used to study the behaviors of young preschool children aged from 1.5 to 5 years and school-age children and adolescents aged from 6 to 18 years. The tool can be used in therapeutic, medical, educational and legal settings to measure adaptive behavior and emotional issues. The test is completed by a parent or guardian who has at least fifth-grade reading skills and in-depth knowledge of the child.

CBCL 1.5–5 consists of 99 items that concern different behavioral areas of the child. When answering the questions, parents should refer to the current status of their child’s life and that of the last two months of their child’s life. In addition, they should rate each item according to the intensity and frequency that best describes their child:-Score 0: the item is not true;-Score 1: the item is somewhat or sometimes true;-Score 2: the item is very true or often true.

The scoring of the questionnaire provides three profiles:The main scales:
-Internalizing;-Externalizing;-Total problems.
The syndromic scales:
-Emotionally Reactive;-Anxious/Depressed;-Somatic Complaints;-Withdrawn/Depressed;-Attention Problems;-Aggressive Behavior;-Sleep Problems.
DSM-oriented scales:
-Affective Problems;-Anxiety Problems;-Pervasive Developmental Problems;-Attention Deficit/Hyperactivity Problems;-Stress Problems;-Autism Spectrum Problems;-Oppositional Defiant Problems.


These profiles allow for the determination of whether scores are in the normal, borderline, or clinical range:N: normal range;N–B: borderline clinical range;N–C: clinical range.

CBCL 6–18 consists of 113 items, and scoring is assigned using a 3-point Likert scale:0 = “absent”;1 = “occurs sometimes”;2 = “occurs often”.
The 2001 revised version of the CBCL is structured around 8 syndromic scales:Anxious/depressed;2.Withdrawn/depressed;3.Somatic complaints;4.Social problems;5.Thought problems;6.Attention problems;7.Rule-breaking behavior;8.Aggressive behavior.Syndromes are further combined into:-Internalizing: 1, 2, 3;-Externalizing: 7, 8;-Total problems.The 2021 revision of CBCL has a scale showing scores associated with disorders from the Diagnostic and Statistical Manual of Mental Disorders (DSM-IV-TR)-Affective disorders;-Anxiety disorders;-Somatic problems;-Attention deficit hyperactivity disorder (ADHD);-Oppositional defiant disorder;-Conduct problems.

The collected data can contribute to diagnosis in several ways. The test correction allows for a score to be reported, the T score, which identifies the result as:N: normal range;N–B: borderline clinical range;N–C: clinical range.

#### 2.1.2. SDSC: Sleep Disturbance Scale for Children

The SDSC, which consists of 26 Likert-type items, was created to assess different types of sleep disturbances in children. A five-point Likert-style scale is used by parents to indicate how often their children exhibit particular behaviors:

1 = never;

2 = occasionally (once or twice a month or less);

3 = sometimes (once or twice a week);

4 = often (3 to 5 times a week);

5 = every day.

The sum of the items identifies a score related to each of the six types of disorders investigated and an overall questionnaire score:DIMS: disorders of initiating and maintaining sleep (sum the score of the items 1, 2, 3, 4, 5, 10, 11). DIMS is clinically significant if the score is superior to 17.SBD: sleep breathing disorders (sum the score of the items 13, 14, 15). SBD is clinically significant if the score is superior to 7.DA: disorders of arousal (sum the score of the items 17, 20, 21). DA is clinically significant if the score is superior to 6.SWDT: sleep–wake transition disorders (sum the score of the items 6, 7, 8, 12, 18, 19). SWDT is clinically significant if the score is superior to 14.DOES: disorders of excessive somnolence (sum the score of the items 22, 23, 24, 25, 26). DOES is clinically significant if the score is superior to 13.SHY: sleep hyperhidrosis (sum the score of the items 9, 16). SHY is clinically significant if the score is superior to 7.Total score: clinically significant if the score is superior to 71.

An overall score ≥ 71 indicates a pathological sleep profile for the presence of at least one disorder.

### 2.2. Statistical Analysis

Patients’ demographic characteristics and questionnaire scores in the case and control group were defined using descriptive statistics and expressed as a percentage. Subsequently, we compared the questionnaire scores between the two groups using the chi-square test. A significance level of *p* < 0.05 was set to determine statistical significance. All analyses were performed using Microsoft Excel for Microsoft 365, Microsoft Inc, Redmond, WA, USA, and IBM SPSS Statistics for Windows, Version 26.0, Armonk, NY, USA: IBM Corp.

## 3. Results

### 3.1. Results of the Whole COVID-19 Group

A total of 107 patients were included in the study. The mean age of the COVID-19 study group was 9.4 ± 4.6 years, with a male prevalence of 55.7%. From the analysis of the SDSC questionnaire, the prevalence of clinically significant values was determined for each item: DIMS: 11.3%; SBD: 17%; DA: 10.4%; SWDT: 14.2%; DOES: 2.8%; SHY: 11.3%; total score: 1.9% (Figure 1). The major prevalence was found for the SBD scale (17%).

The results that emerged from the analysis of the CBCL questionnaire show the highest prevalence of clinically significant and borderline outcomes, respectively, for the following items: 20.8% and 9.4% for the internalizing scale, 9.4% and 14.2% for the total scale, 5.66% and 17.9% for anxiety/depression, 4.7% and 11.3% for the externalizing scale, 5.6% and 12.2% for withdrawal, 10.4% and 4.6% for somatic complaints, 7.1% and 5.8% for thought problems (Figure 2). 

#### 3.1.1. 1.5–5-Year-Old Sample

The sample included 21 children aged between 1.5 and 5 years, with an average age of 3.2 ± 1.4 years and a male prevalence of 57.1%. The percentage frequencies of every CBCL questionnaire item were evaluated for clinical, borderline and normal results. 

We found a prevalence of 4.8% for borderline results for the following items: emotional reactivity, externalizing scale, somatic complaints, affective disorders, ADHD and total score. A prevalence of 5% of borderline results was observed for ‘attention problems’ item. For the item ‘thought disorders’, we reported a prevalence of borderline and clinical results at 5.9% and 1.2%, respectively. For ‘sleep problems’, we found only clinical results, with a prevalence of 5% (Figure 3). 

The correlation between sleep and behavioral disorders was also evaluated (Table 2). Statistically significant results were found for the correlation between the total score of CBCL, SWDT and DOES (*p* < 0.001, r 0.58 and *p* < 0.05, r 0.50 respectively). Additionally, statistically significant results were found for the correlation between the externalizing scale of CBCL, SWDT and DOES (*p* < 0.001, r 0.58 and *p* < 0.05, r 0.50, respectively).

#### 3.1.2. 6–18-Year-Old Sample

The study included 85 children aged between 6 and 18 years; the average age of the population was 11.3 ± 3.4 years, with a male prevalence of 55.3%. For each item of the CBCL questionnaire, the percentage frequencies were evaluated for clinical, borderline and normal results. The highest prevalence of clinically significant and borderline outcomes was observed for the following items: anxiety/depression 16.5%, 8.2%; withdrawal 12.9%, 5.9%; somatic complaints 4.7%, 12.9; internalizing scale 11.8%, 25.9%; externalizing scale 12.9%, 5.9; total score 17.7%, 11.7; anxiety disorders: 11.8, 17.7%; somatic problems 13.1%, 7.1% (Figure 4 and Figure 5). 

The correlation between sleep and behavioral disorders was also evaluated in this group of patients. Statistically significant results were observed for the correlation between total score, internalizing and externalizing scale and SDSC questionnaire items (Table 3). 

The frequency distribution of behavioral disorders in pediatric age groups (6–18 and 1–5) showed the following statistically significant results: anxiety/depression: *p* < 0.05; internalizing scale *p* < 0.01; total score *p* < 0.05; anxiety disorders *p* < 0.05.

### 3.2. Control Group: Prevalence of Disorders and Comparison to COVID-19 Patients

The control group consisted of 100 patients, aged from 1.5 to 18 years, in a period prior to the COVID-19 pandemic. The mean age was 9.6 ± 2.2 and the male prevalence was 46.9%. We determined the frequency of sleep disorders: 5.1% for DIMS, 10.2% for SBD, 7.1% for DA, 10.2% for SWDT, 5.1% for DOES, 4.1% for SHY, 0% for total score (Figure 6). 

The analysis of the prevalence of behavioral disorders showed the greatest results for internalizing problems (14.3%), social problems (7.1%) and total score (7.1%). (Figure 7).

The comparison of CBCL results between cases and controls revealed statistically significant differences for the scales/items: anxious/depressed problems (*p* < 0.001), withdrawn/depressed (*p* < 0.001), somatic complaints (*p* < 0.01), thought problems (*p* < 0.01), internalizing problems (*p* < 0.01), externalizing problems (*p* < 0.01) and total score (*p* < 0.01) (Figure 8). 

However, the comparison of SDSC results between cases and controls showed no significant difference per any items. IPN was the item that showed the closest to a statistically significant difference (*p* = 0.069).

## 4. Discussion

Several investigations revealed that neuropsychiatric disorders are the most common and persistent manifestations of long COVID in children [18,22,41]. In our study, we assessed the prevalence of sleep and behavioral disorders using two validated scales (CBCL and SDSC) in children aged 1.5–18 years who contracted COVID-19 between one year and one month prior to data collection, and compared the results with a control group (pre-COVID-19 pandemic. In the COVID-19 group, we found the highest prevalence for the item SBD of the SDSC questionnaire and for the items internalizing, anxiety/depression and total score of the CBCL questionnaire. The analysis of the sample divided by age showed that, in the sample aged 6–18 years, there were more results in the borderline and clinically significant range for the total scale, withdrawal and anxiety/depression. The major prevalence for clinically significant values was found for the internalizing scale. For the syndromic scales of DSM-oriented disorders, a prevalence of 17.7% was found for anxiety disorders in the clinically significant range, which increased to 29.5% if results for the borderline range were included. An interesting result was also found for the main scales of CBCL (total, internalizing and externalizing scales) compared with the sleep disturbance scales of SDSC. Statistically significant results were obtained for the correlation between total scale with sleep initiation and maintenance disorders (DIMS), sleep breathing disorders (SBD), arousal disorders (DA), sleep–wake transition disorders (SWDT) and the total score. The correlation between the externalizing scale and sleep disorders was statistically significant for every item except for arousal disorders (DA); instead, for the internalizing scale, a statistically significant correlation was observed for every SDSC item except for sleep-breathing disorders (SBD). In the subgroup of patients 1.5–5, no clinically significant results were found, partly certainly due to the smallness of the sample (21 patients). However, a statistically significant correlation was found between total score and externalizing scale and SDSC items, DOES and SWDT. The comparison between COVID-19 patients (cases) and the pre-pandemic group (controls) revealed statistically significant differences, particularly for the internalizing scale (*p* < 0.01), which included anxious/depressive symptoms, withdrawal and somatic complaints, with a prevalence of 30.2% compared to 14.3% in the control sample. Also, for the total scale, we found a remarkable result of 23.6% compared to the controls, in which the frequency was 7.1%. Moreover, the anxiety/depression item reached a statistically significant result, with an increase of 23.6% in COVID-19 patients compared to 5.1% of controls. However, the comparison of SDSC results between cases and controls showed no significant difference for any items. These preliminary results show that COVID-19 pandemic had a significant impact on children and adolescents’ mental health, particularly in adolescents, where internalizing disorders, especially anxiety/depression problems, recorded the highest results. During the pandemic, in fact, there was an increase in mental emergency consultations for teenagers [42]. In the Italian study carried out by Guido et al., among 322 COVID-19 patients aged between 1.5 and 17 years old, neuropsychological symptoms were analyzed at onset, after 1 month, and after 3–5 months using standardized questionnaires including CBCL. Twenty-two percent of patients showed symptom persistence 3–5 months after disease onset. The most common long-COVID neurological symptoms were headache, fatigue, and anosmia [43]. In both the 1.5–5-year-old and 6–18-year-old subgroups, internalizing problems prevailed over externalisation problems. In the 6–18-year-old subgroup, anxiety (n = 74, 28%), depression (n = 51, 19%), somatic problems (n = 43, 16%), attention problems (n = 20, 8%), oppositional problems (n = 16, 6 %), and conducted (n = 6, 2%) were the most prevalent disorders [44]. In the research by Jarvers et al., internalizing and externalizing problems significantly increased with time in preschoolers, peaking during the lockdown and only slightly declining after the lockdown [45]. Schmidt et al. found that, among 5823 children, from 2.2% to 9.9% reported emotional and behavioral issues that were beyond the clinical cut-off. Adolescents reported the greatest increase in emotional problems, whereas preschoolers (ages 1–6) showed the greatest increase in oppositional–defiant behaviours [46]. Furthermore, sleep disturbances related to the pandemic were demonstrated by many studies [32,47,48,49]. Sleep habits changed during the pandemic, particularly during lockdown. School-age children and adolescents were going to bed later and waking later, with disturbed sleep quality [50]. In the research by Liu et al., conducted in preschoolers, a higher prevalence of sleep disturbances during lockdown was shown in 2020 than in the control group [51]. The study by Wong et al., exploring the impact of the COVID-19 pandemic on the sleep–wake patterns of preschool children revealed altered sleep–wake patterns [52]. Nearly half of the 129 Italian children with COVID-19 who participated in the study by Buonsenso et al. reported at least one symptom 60 days or more after SARS-CoV-2 infection, including fatigue, soreness in the muscles and joints, headaches, sleeplessness and problems concentrating. Even most children with asymptomatic COVID-19 developed chronic, persistent symptoms [22]. There are few studies in the literature that explored the neuropsychological profile of pediatric patients who had COVID-19 using CBCL and SDSC questionnaires as assessment tools. The strengths of this study are the comparison with a control group and the use of standardized and replicable tools. Our study has also some limitations: the sample of patients was relatively small; clinical features of both groups are missing; there are few studies similar to ours using standardized psychological scales (CBCL and SDSC) on paediatric patients with long COVID, which aspect prevents us from conducting a systematic comparison of symptoms with clinical samples that are comparable to ours; data were not collected in the following months to assess the evolution of symptoms.

## 5. Conclusions

Mental health at pediatric age is a source of constant concern for clinicians. The impact of the COVID-19 pandemic on mental health is undeniable and our study confirms a sharp increase in behavioral disorders in pediatric patients, above all in adolescents, compared to the control cohort in the pre-pandemic era. Further studies are needed to assess whether such disorders can last for months, or even continue into adulthood. Surely, among the ‘lessons’ of COVID-19, the need to take care of mental health is also revealed. Carrying those lessons forward into the “new normal” may help us to remember to be proactive about caring for our psychological well-being.

## Figures and Tables

**Figure 1 children-10-01189-f001:**
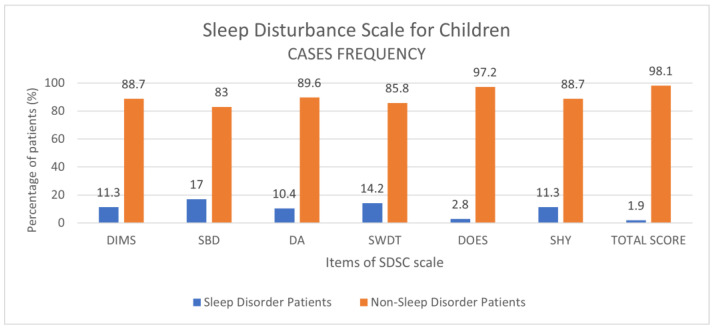
SDSC item results, expressed as percentage frequencies in the sample of 107 COVID-19 patients.

**Figure 2 children-10-01189-f002:**
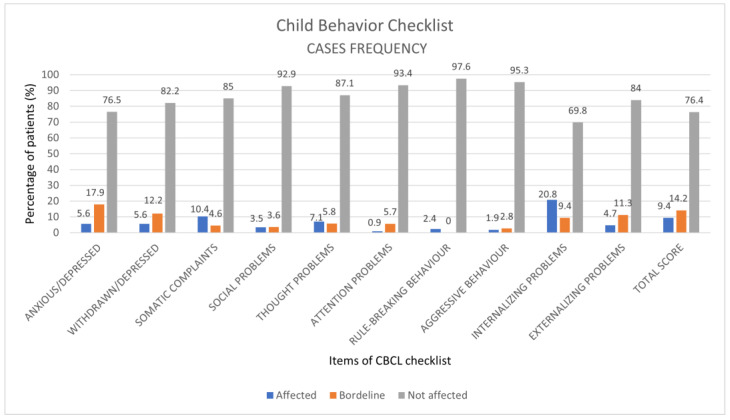
CBCL item results, expressed as percentage frequencies, for syndromic scales divided into clinical, borderline and normal ranges in the sample of 107 COVID-19 patients.

**Figure 3 children-10-01189-f003:**
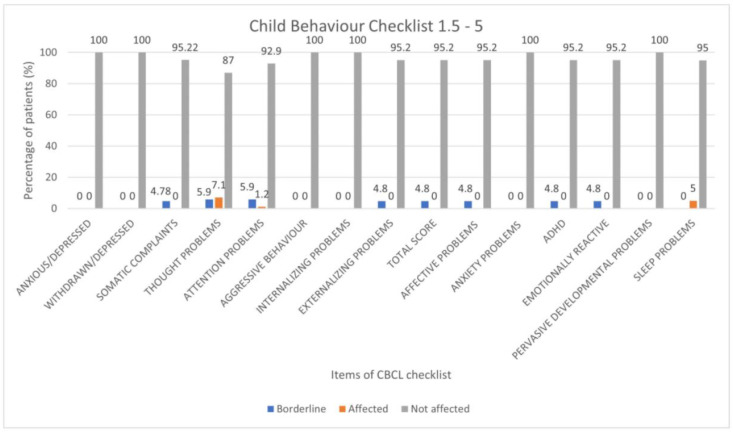
Results of CBCL scales 1.5–5 expressed as categorical variables (percentage frequencies).

**Figure 4 children-10-01189-f004:**
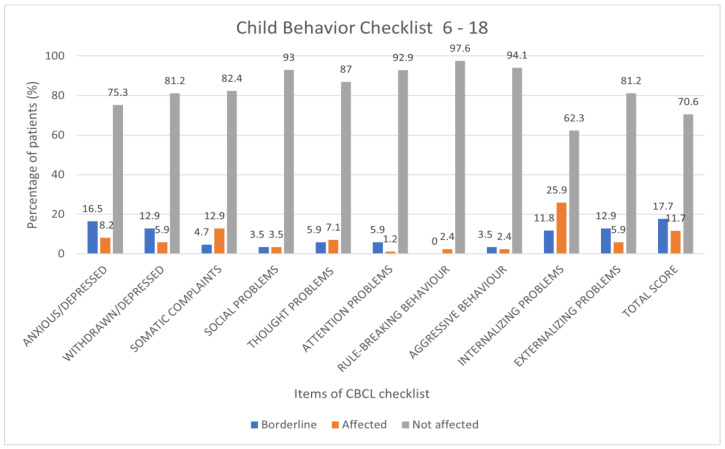
Results of CBCL scale 6–18 expressed as categorical variables (percentage frequencies).

**Figure 5 children-10-01189-f005:**
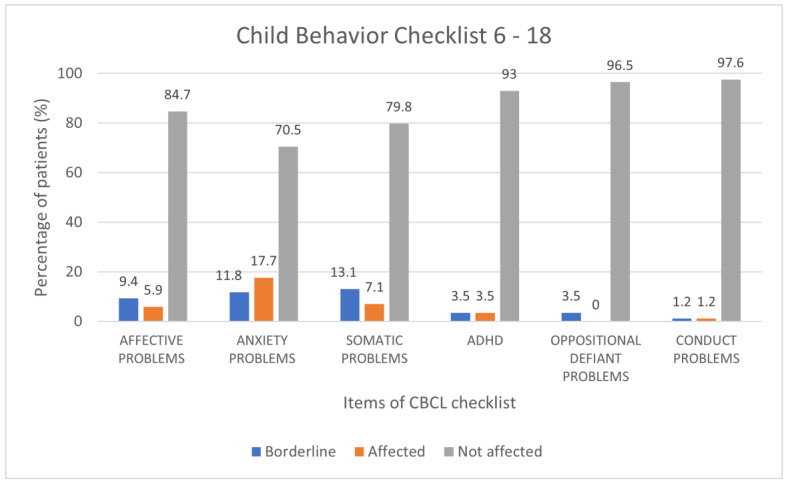
Results of the CBCL 6–18 scales expressed as categorical variables (percentage frequencies).

**Figure 6 children-10-01189-f006:**
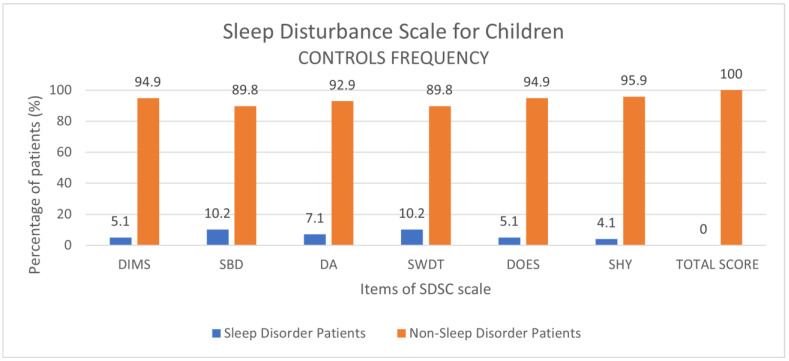
Results of the SDSC scale expressed as categorical variables (percentage frequencies) in the control group.

**Figure 7 children-10-01189-f007:**
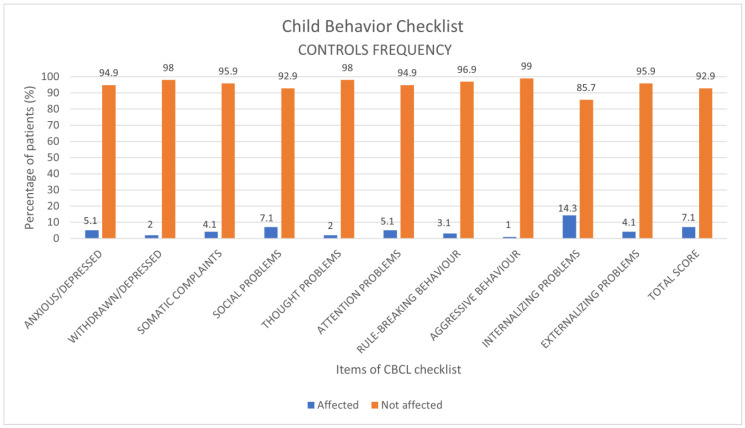
Results of the CBCL scale expressed as categorical variables (percentage frequencies) in the control group.

**Figure 8 children-10-01189-f008:**
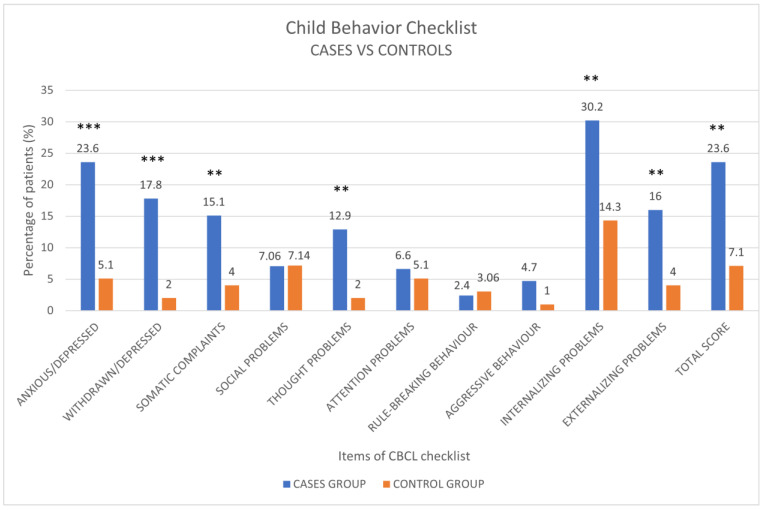
Results of the comparison between case and control for CBCL items. ** statistically significant (*p* < 0.01), *** statistically significant (*p* < 0.001).

**Table 1 children-10-01189-t001:** Definition of long COVID or post-COVID syndrome.

Definition of Long COVID or Post-COVID Syndrome	
National Institute for Clinical Excellence (NICE) [13]	Symptoms of COVID-19 experienced for 12 or more weeks after initial recovery. Ongoing symptomatic COVID-19: signs and symptoms that persist from 4 to 12 weeks after acute COVID-19; Post-COVID syndrome: signs and symptoms that develop during or after an infection consistent with COVID-19, continue for more than 12 weeks and are not explained by an alternate diagnosis.
Centers for Disease Control and Prevention (CDC) [14]	The post-COVID condition indicates consequences that are present >4 weeks after SARS-CoV-2 infection. This includes both general complications of prolonged acute illness and new, returning, or ongoing health problems as post-acute sequelae of SARS-CoV-2 infection (PASC).
Robert Koch Institute (RKI) [15]	Long COVID is a longer-term health impairment following a SARS-CoV-2 infection that is present beyond the acute phase of the sickness of 4 weeks. The symptoms either begin during the acute phase of the disease and persist for a longer period of time, or appear or reoccur in the course of weeks and months after the infection.Post-COVID condition or post-COVID syndrome: symptoms are either still present at least 12 weeks and longer after the acute infection, or appear anew after this period and cannot be explained otherwise.

**Table 2 children-10-01189-t002:** Correlation between the total CBCL scale and the CBCL externalizing scale and the SDSC syndromic scales.

	Total Score		Externalizing Scale	
SDSC Scale	*p*	*R*	*p*	*R*
SWDT	<0.001	0.58	<0.001	0.59
DOES	<0.05	0.50	<0.05	0.53

**Table 3 children-10-01189-t003:** Correlation between the total, externalizing and internalizing CBCL scale and the SDSC syndromic scales: statistically significant results.

	Total Score		Externalizing		Internalizing	
SDSC	*p*	*R*	*p*	*R*	*p*	*R*
DIMS	<0.01	0.31	<0.01	0.34	<0.05	0.23
SBD	<0.05	0.23	<0.05	0.23	/	/
DA	<0.05	0.24	/	/	<0.05	0.23
SWDT	<0.001	0.35	=0.001	0.34	<0.01	0.31
DOES	<0.01	0.39	<0.05	0.24	<0.05	0.24
TOTAL	<0.001	0.37	<0.05	0.33	<0.01	0.30

## Data Availability

The datasets used and/or analyzed during the current study are available from the corresponding author on reasonable request.

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
