# Peer review of "Behavioral and Sleep Disorders in Children and Adolescents following COVID-19 Disease: A Case-Control Study"

_children, 2023, doi:10.3390/children10071189_

Round 1

Reviewer 1 Report

The prevalence of post-COVID disorders has been of great interest recently, due to the large number of patients who have had a coronavirus infection and have complaints over the next year. Most often, this problem concerns adults who have recovered from covid-19, this is due to the fact that coronavirus infection in children is milder and the compensatory mechanisms of the child's body are higher than in adults. However, despite this, symptoms of post-cavid disorders also occur in the middle and older children's age group. In this regard, the topic of this original study is relevant and interesting.

A few things came up during the review process:

1) The authors need to structure the abstract according to the points highlighted in the article (introduction, purpose, materials and methods ...)

2) There is no reference in the text to Table 1. For clarity, it is recommended that this table be converted to a block diagram.

3) Item Materials and methods: the authors need to add the clinical characteristics of the observed groups (age, gender, somatic pathology, duration of covid-19, severity of the coronavirus infection, information about whether the children received / are receiving medications aimed at combating post-covid symptoms, the presence of an additional stress factor in the family), it is recommended to present the results in the form of a table. It is also necessary to prescribe inclusion criteria and exclusion criteria from the study.

4) The numbering of sections is completed by paragraph 2 - Materials and methods, it is also necessary to number the following sections of the article (statistical analysis, results, discussion, conclusion)

5) In the figures, it is necessary to indicate the units of measurement and sign the axes.

6) Due to the fact that there is not enough information in separate tables, it makes sense to combine tables: 2 and 3, as well as tables: 4, 5, 6 among themselves.

Author Response

1) The authors need to structure the abstract according to the points highlighted in the article (introduction,
purpose, materials and methods ...) Thank you for the comment, we have edited and highlighted this part.
2) There is no reference in the text to Table 1. For clarity, it is recommended that this table be converted to a
block diagram. Thank you for the comment, we have edited and highlighted this part.

3) Item Materials and methods: the authors need to add the clinical characteristics of the observed groups
(age, gender, somatic pathology, duration of covid-19, severity of the coronavirus infection, information about
whether the children received / are receiving medications aimed at combating post-covid symptoms, the
presence of an additional stress factor in the family), it is recommended to present the results in the form of a table. It is also necessary to prescribe inclusion criteria and exclusion criteria from the study. Thank you for
your comment. Unfortunately, we don’t have information regarding the somatic pathology, duration of COVID-19, etc of the case group. We have reported the inclusion and exclusion criteria from the study in the text.
4) The numbering of sections is completed by paragraph 2 - Materials and methods, it is also necessary to number the following sections of the article (statistical analysis, results, discussion, conclusion) Thank you
for the comment, we have edited and highlighted this part.
5) In the figures, it is necessary to indicate the units of measurement and sign the axes. Thank you for the comment, we have edited and highlighted this part.
6) Due to the fact that there is not enough information in separate tables, it makes sense to combine tables:
2 and 3, as well as tables: 4, 5, 6 among themselves Thank you for the comment, we have edited and highlighted this part.

Reviewer 2 Report

This study addresses the timely topic of behavioral and sleep disorders onset in children and adolescents after COVID-19 disease, leading to conclusions that may be useful for clinicians involved in the case management of these patients. However, some points need to be addressed, please see below:

Line 91- „In some patients with COVID-19 may develop...” is incorrect grammatically, consider rephrasing it as „Some patients with...” or „In several patients with COVID-19.... has been reported”;

Line 120- informed consent was obtained for every patient from his/her parent/caregiver, I assume; please specify this here, in the „Methodology” section;

Line 118- how were the patients screened for previous behavioral and sleep disorders, before the SARS-CoV-2 infection? Any structured method, like an inventory or a questionnaire for parents? Was the same method of screening applied to the study and to the control group?

Line 120- how was the control group formed? Were the subjects controlled for age, gender, and medical background? What does „the same characteristics” mean in this context, since the main inclusion criteria previously mentioned can not be extrapolated to the control group (except for the age criterion)?

Line 123- „Informed consent was obtained for every patient” is repeated twice. Did you mean it was obtained for each patient in the study and control groups? What type of pathology had the patients in the control group? Do any of the diagnoses in the control group may be considered confounders in the comparative analysis with the study group? Can this be a limitation of the study?

Line 225- „COVID-19 total population” is an imprecise formulation, consider replacing this with „The analysis of the whole study group” or „The results of the whole COVID-19 group”;

Fig.1 and 2- on the ordinate of these graphs are represented percentages, if I understood correctly, therefore please define the superior limit as 100; also, in fig.2, isn’t the sum for the three values, i.e. affected, borderline and non-affected, for each variable, to be 100%? It seems counterintuitive that, in the same group, 85% of the subjects had no somatic complaints, 15.1% were borderline and 10.4% had no such manifestations...

The same observation about the ordinate values’ superior limit applies to all the graphs.

Line 353- it is recommendable to avoid beginning a sentence with a numeral, otherwise consider spelling it out;

Any suggestions for further studies in this field?

Minor English language errors have been identified.

Author Response

Line 91- „In some patients with COVID-19 may develop...” is incorrect grammatically, consider rephrasing it
as „Some patients with...” or „In several patients with COVID-19.... has been reported”;  Thank you for the
comment, we have edited and highlighted this part.
Line 120- informed consent was obtained for every patient from his/her parent/caregiver, I assume; please
specify this here, in the „Methodology” section; Thank you for the comment, we have edited and highlighted this part.
Line 118- how were the patients screened for previous behavioral and sleep disorders, before the SARS- CoV-2 infection? Any structured method, like an inventory or a questionnaire for parents? Was the same
method of screening applied to the study and to the control group? the control group is a group that shares the same age, gender as the case group, as well as the absence of neuropsychiatric disorders’ diagnosis. The parents of the control group filled in the same questionnaires (CBCL and SDSC). The group is antecedent to SARS-CoV-2 infection. The same screening method was applied to both groups.
Line 120- how was the control group formed? Were the subjects controlled for age, gender, and medical
background? What does „the same characteristics” mean in this context, since the main inclusion criteria
previously mentioned can not be extrapolated to the control group (except for the age criterion)? the control group is a group that shares the same age, gender as the case group, as well as the absence of neuropsychiatric disorders’ diagnosis. We added the mean age and the sex prevalence of the control group.
Line 123- „Informed consent was obtained for every patient” is repeated twice. Did you mean it was obtained
for each patient in the study and control groups? What type of pathology had the patients in the control
group? Do any of the diagnoses in the control group may be considered confounders in the comparative
analysis with the study group? Can this be a limitation of the study? Thank you for the comment, we have edited and highlighted this part. The informed consent was obtained for each patient in the study and control groups. Unfortunately, we don’t have information regarding the pathologies of the control group. We added this aspect as a limitation of the study.
Line 225- „COVID-19 total population” is an imprecise formulation, consider replacing this with „The analysis
of the whole study group” or „The results of the whole COVID-19 group”; Thank you for the comment, we have edited and highlighted this part.
Fig.1 and 2- on the ordinate of these graphs are represented percentages, if I understood correctly, therefore
please define the superior limit as 100; also, in fig.2, isn’t the sum for the three values, i.e. affected,
borderline and non-affected, for each variable, to be 100%? It seems counterintuitive that, in the same
group, 85% of the subjects had no somatic complaints, 15.1% were borderline and 10.4% had no such
manifestations...
The same observation about the ordinate values’ superior limit applies to all the graphs. Thank you for the comment, we have edited and highlighted this part in the text and in the figures.
Line 353- it is recommendable to avoid beginning a sentence with a numeral, otherwise consider spelling it
out; Thank you for the comment, we have edited and highlighted this part.
Any suggestions for further studies in this field? Thank you for the question. We think that further studies are needed to assess whether such disorders can last for months or even continue into adulthood and the associated risk factors.

Reviewer 3 Report

Hi Dears;

Thanks for the excellent research, especially in the field of children and COVID 19

In my opinion ,there is no significant problem in the research.

Author Response

Thank you for the comment.